# Measuring Italian Citizens' Engagement in the First Wave of the COVID-19 pandemic containment measures: A cross-sectional study

**Guendalina Graffigna**[1,2,3], **Serena Barello**[1,2]*, **Mariarosaria Savarese**[1,3], **Lorenzo Palamenghi**[1,3], **Greta Castellini**[1,3], **Andrea Bonanomi**[4], **Edoardo Lozza**[2]

**1** EngageMinds HUB–Consumer, Food & Health Engagement Research Center, Università Cattolica del Sacro Cuore, Milan, Italy, **2** Department of Psychology, Università Cattolica del Sacro Cuore, Milan, Italy, **3** Faculty of Agricultural, Nutrition and Environmental Sciences, Università Cattolica del Sacro Cuore, Milan, Italy, **4** Department of Statistical Science, Università Cattolica del Sacro Cuore, Milan, Italy

* serena.barello@unicatt.it

## Abstract

### Background

In January 2020, the coronavirus disease 2019 (COVID-19) started to spread in Italy. The Italian government adopted urgent measures to slow its spread. Enforcing compliance with such measures is crucial in order to enhance their effectiveness. Engaging citizens in the COVID-19 preventive process is urgent today both in Italy and around the world. However, to the best of our knowledge, no previous studies have investigated the role of health engagement in predicting citizens' compliance with health emergency containment measures.

### Method

An online survey was administered between February 28 and March 4, 2020 on a representative sample of 1000 Italians. The questionnaire included a measure of health engagement (Patient Health Engagement Scale), a 5-item Likert scale ranging from 1 to 7, resulting in 4 positions that describe the psychological readiness to be active in one's own health management, and a series of ad hoc items intended to measure citizens' perceived susceptibility and severity of the disease, orientation towards health management, trust in institutional bodies, health habits and food consumption. To investigate the relationship between health engagement and these variables, ANOVA analysis, logistic regression and contingency tables with Pearson's chi-squared analysis have been carried out.

### Results

Less engaged people show higher levels of perceived susceptibility to the virus and severity of the disease; they are less trustful of scientific and healthcare authorities, they feel less self-effective in managing their own health—both in normal conditions and under stress—

**Data Availability Statement:** All relevant data are within the manuscript and its Supporting Information files.

**Funding:** This study was conducted within the CRAFT project, funded by Fondazione Cariplo & Regione Lombardia.

**Competing interests:** The authors have declared thata no competing interests exist.

and are less prone to cooperate with healthcare professionals. Low levels of health engagement also are associated with a change in the usual purchase behavior.

## Conclusions

The Patient Health Engagement model (PHE) provides a useful framework for understanding how people will respond to health threats such as pandemics. Therefore, intervention studies should focus on raising their levels of engagement to increase the effectiveness of educational initiatives intended to promote preventive behaviors.

## Introduction

In January 2020, the coronavirus disease 2019 (COVID-19) started to spread in Italy. The virus and its associated disease were given the designation of coronavirus disease 2019 (COVID-19) in February 2020, distinguishing this syndrome from the acute respiratory syndromes associated with 2 other betacoronaviruses (SARS-CoV and Middle East respiratory syndrome corona-virus) that caused earlier outbreaks of severe disease [1,2]. As of March 17, 2020, a total of 31,506 COVID-19 cases with 2503 deaths and 2941 recovered had been reported in Italy (updated Italian situation available at http://www.salute.gov.it/portale/nuovocoronavirus/dettaglioContenutiNuovoCoronavirus.jsp?area=nuovoCoronavirus&id=5351&lingua=italiano&menu=vuoto).

On Jan 30, 2020, the World Health Organization (WHO) declared the coronavirus outbreak a public health emergency of international concern (PHEIC). Starting March 7, 2020, the Italian government adopted very urgent and restrictive measures to slow the virus spread and reduce its potential impact on the population (available at https://www.gazzettaufficiale.it/eli/id/2020/03/08/20A01522/sg). Several cities—identified as "red areas"—have been put under quarantine, hoping to stop the disease from spreading to other parts of the country. This situation is globally unprecedented at least for two main reasons. First, to control the COVID-19 outbreak, governmental authorities have suddenly adopted very extreme public measures such as locking down cities, deeply reorganizing healthcare services to cope with the rapid increasing demand for acute care, imposing school and university closures, suggesting—where possible—smart-working solutions and transportation restrictions, deploying thousands of healthcare workers to more heavily affected regions, and running wide public health messaging campaigns for consumers' education. Second, consumers are overwhelmed by rather mixed and confounding information, partly because scientific discovery related to COVID-19 disease is constantly evolving with the course of the disease outbreak, and partly due to the rapid increase in misleading or false news. Therefore, all these measures are currently having a deep impact on Italian people's attitudes, daily habits and consumption behaviors [3].

As in other similar situations, prior to the availability of an effective vaccination therapy, strategies to mitigate and control the impact of the pandemic typically involve "non pharmacological" interventions [4,5], and rely on citizens' autonomous responses to public health preventive measures. In particular, past literature suggested that people appear to respond to an epidemic by voluntarily undertaking specific behaviors in order to protect themselves [6,7]. However, in some cases, these behaviors may not correspond to an objective evaluation of risk [8,9], but depend on individual subjective evaluations, thus becoming potentially counterproductive. For this reason, there has been a rapid rise of interest in understanding the

determinants of people's behavioral change that may influence the adequacy of the response to health emergencies [10,11]. People dealing with these situations, indeed, may experience negative attitudes, feelings of uncertainties and alarmism [12]. These reactions might potentially end in risky habits and inadequate and disorganized behaviors, both for individuals and the community [13,14], affecting public health outcomes. Therefore, to study the subjective factors implied in such reactions is of much relevance to effectively sensitize the general public and identify high-risk targets [15]. Along with structural and immutable factors such as socio-demographics, scholars have previously attempted to understand the subjective determinants of citizens' changing attitudes and behaviors in a pandemic emergency. In particular, authors identified risk perception as one of the most relevant variables in determining citizens' response to global pandemic disease [16,17], among a series of factors such as the perception of economic impact [18]; efficacy beliefs related to health [19]; level of literacy and knowledge elaboration [20,21]. Another important factor identified is the level of subjective anxiety, which influences both citizens' attitudes towards the emergency disease and consequent preventive behaviors [22–24]. Other subjective factors accounting for the changes in peoples' habits in pandemic emergencies include those related to the perceived effect of one's individual behavior, such as perceived costs and benefits of preventive behaviors on the disease spread [25,26] or perceived impact of an individual's behavior on other individuals' outcomes [24,27].

Among other variables accounting for a change in citizens' attitudes, habits and behaviors, scholars recently have shown the role of health engagement in affecting health-related behaviors and preventive habits [28–34]. More specifically, people with high levels of health engagement have been identified as more likely to adopt behavioral change suggestions and to adhere to medical prescriptions [35–39]. However, previous literature has also demonstrated how individuals may be in different phases of their process of engagement [29,31,40,41], thus being more or less ready to enact a change in the way they cope with a critical event and comply with prescribed preventive conduct [42–45].

In the current COVID-19 outbreak in Italy–as well as in other countries–citizens are experimenting with a life-threatening situation, leading to profound changes in attitudes, habits and behaviors, which also are potentially negative for consumers' health and virus containment. Making citizens aware of their crucial role in avoiding the rapid spread of the virus and engaging them in the COVID-19 disease preventive process is urgent today both in Italy and around the world.

However, to the best of our knowledge, no previous studies have investigated the role of people's health engagement in determining citizen attitudes, habits and compliance with containment measures in the event of an health emergency; moreover, previous literature has highlighted the need to apply a validated theoretical framework to the study of these phenomena in order to effectively predict people's responses to the event and adherence to prescriptions [11].

For these reasons, we conducted a study aimed at understanding citizens' attitudes and behavioral responses to the current spread of COVID-19 disease in Italy and determining how they changed their daily habits and behaviors according to their level of health engagement. Results of this study will contribute to informing public health communication and targeted consumer education activities.

## Theoretical background

The Patient Health Engagement model [29,46] is a recently developed psychological framework that theorizes how individual health engagement results from a continuous emotional and motivational reframing of an individual's own role perception in the management of a

disease (i.e. from passive user of services to active partner of the healthcare system). According to this model, to become engaged means to be emotionally resilient and able to adjust to the health risks and specific requirements. This model also features unique ways of coping with a health crisis, which the Covid-19 disease can be considered today, with the necessary adaptation to the specific context. In particular, the model features four positions: the first position ("Blackout") is complete disengagement, typically occurring when people feel vulnerable and without control over the perceived risk, psychologically frozen and behaviorally paralyzed. In the case of COVID-19 disease, this position can explain the psychological reaction of all the citizens to the very initial moment of pandemic spread, where the sudden recognition of the uncontrollable disease diffusion changed people's lives in the most hit countries. Moreover, the sense of helplessness also exposed the citizens to a psychological vulnerability and risk. Next follows the psychological position of "Arousal," in which people have acquired an initial awareness about their actual situation of health risk but don't yet have enough knowledge and skills to manage it. They do not accept the impact of preventive requirements on the modification of their daily habits and appear hyper-vigilant over their body signals, disorganized and confused when seeking information on the health situation. In the situation here described, each unexpected news or change in the epidemic situation causes emotional alert and overwhelming emotional response, with disorganized actions and behaviors that can be dysfunctional for health prevention. When individuals succeed in the process of emotional regulation and coping with the stressful condition, they achieve a position of "Adhesion." In this phase, patients have developed a good psychological adaptation to the critical situation and appear able to manage their psychological distress connected–in this case—to the COVID-19 health emergency. They appear more motivated to comply with medical and preventive prescriptions. In this phase, moreover, patients acquire further skills to effectively manage their risk condition. Finally, when people achieve a complete awareness of the characteristics and consequences of the critical situation, and assume a more responsible position in their behaviors and risk management, they reach the "Eudaimonic project" phase, which features a better, positive and optimistic approach to the situation, with an increased ability to deal with the uncertainty of the moment and a strong motivation to psychologically achieve the sense of a "new normality" (Fig 1), to overcome the current emergency. The model has been translated into a psychometric scale used to measure Patient Health Engagement level (PHE-S), which has been validated in different countries [47–49]. The PHE-S also has been cited by different research and clinical groups [34,50,51].

## Methods

### Study design and participants

The study here described is a part of a broader project (entitled: "*Italian citizens' food habits monitoring from a consumer psychology perspective*") aimed at monitoring Italian consumers' habits. In particular, food consumption behaviors, health behaviors, information gathering and trust towards institutions are tracked over time. It is delivered through online surveys repeated and adjusted over time to track changes in consumers' orientations in relation to the evolving socio-economic situation of the country. In this case, we adapted the survey to explore people's reactions to the COVID-19 disease emergency and how different levels of health engagement correspond to unique patterns of behaviours. The survey is based on a cross-sectional design and was carried out between February 28 and March 4, 2020. A panel provider company. Norstat s.r.l. (https://norstat.it/), was in charge of the participants' selection through stratified random sampling, a sampling method that splits the population in smaller groups (strata) based on sociodemographic characteristics and then samples from these

*DISENGAGEMENT*                                                        *FULL ENGAGEMENT*

**BLACKOUT**

When people feel vulnerable and without control over the perceived risks, psychologically frozen and behaviourally freezed.

**AROUSAL**

When people appear hyper-vigilant over they body signals, confused when seeking health information and activated in a disorganized and maladaptive way.

**ADHESION**

When people start coping with the stressful condition and develop a form of psychological adaptation.

**EUDAIMONIC PROJECT**

When people develop the ability to deal with the uncertainty related to their condition and show strong motivation to achieve a sense of «new normality»

**Fig 1. The patient health engagement model.**

subgroups in order to obtain a sample with the same proportions of, for instance, genders as in the general population [52]. To become part of the panel, people are usually first contacted using random digit dialing that is a technique for drawing a sample of households from the frame or set of telephone numbers. Respondents were rewarded through the Norstat system. A sample of 1000 Italians was involved and weighed to be representative of the Italian population for gender, age, employment, geographical area and dimension of urban center of residence, residents from all the different regions of Italy. To be included in the survey, participants are over 18 years old, are able to read and understand Italian and live in Italy. The percentages relating to the Italian population were taken from the website of ISTAT[53]. People belonging to the online panel were carefully screened for authenticity and legitimacy via digital finger-print and geo-IP-validation from the panel provider. In this study, in order to guarantee data quality, respondents were asked to confirm their demographics. From the 1000 recruited subjects, 32 were excluded because demographic data provided by the respondent and those provided from the panel were inconsistent (there were discrepancies between reported and known gender and/or age). Statistical analyses were hence carried out on a dataset composed of the answers of 968 respondents. All analyses have been carried out with IBM SPSS 23 (release 23.0.0.0).

### Ethical statement

Each participant was instructed about the aims of the research and gave written informed consent before starting the questionnaire. By agreeing to start the compilation, participants accepted the informed consent. They were also allowed to drop out from the compilation at any time. As a part of a panel, the GDPR compliance for the participants here involved is guaranteed by Norstat s.r.l. We received the anonymous database for analysis. No participants' identification detail was provided to researchers. This study has been performed in accordance

with the Declaration of Helsinki and has been approved by an independent ethics committee of Università Cattolica del Sacro Cuore in Milan (CERPS—IRB#02–20).

## Study measures

As a part of a broader study (see section 2.1), the survey is composed of a core of fixed measures and a pull of ad-hoc items that change according to the contingency occurring during the specific data collection period. In the wave of data collection we report on in this article, ad hoc items related to individuals' affective and behavioral responses to the COVID-19 pandemic were added. Following are the specific measures:

- **Health engagement**: we adopted a revised version of the Patient Health Engagement Scale (PHE-s®) to measure this construct. This measure, developed according to the Patient Health Engagement model [46], assesses the consumers health engagement level, defined as the "*people's psychological readiness and sense of mastery to become active players in their own health management and health risk prevention.*" Previous studies demonstrated its robust psychometric proprieties [29], also in other languages [47–49]. This scale features five ordinal items reflecting the continuum of engagement described in the four levels of the PHE model. According to the ordinal nature of the PHE-s®, the median score is considered the more reliable index to calculate the final patients' scoring [29]. According to the score obtained, each respondent result is in one of the four levels of health engagement as described in the PHE model (i.e. blackout, arousal, adhesion, eudaimonic project). The scale is based on the assumption that the score obtained by the person should reflect his/her actual health engagement level. For this study purposes, the PHE- s® was slightly revised in order to adapt the items' formulation to the specific context of the health emergency. The incipit was revised in order to adapt coherently to the Covid-19 emergency (from "thinking about your health" to "thinking about your health in this emergency"; the formulation of the fifth item was also revised to adapt it to the nature of the subject. For this reason, the psychometric characteristics of the revised version were tested.

- **Attitudinal response towards to the COVID-19 health emergency.** In particular, in light of studies on risk processing [54], two elements of risk judgment were measured: (a) risk severity (the perceived potential severity of COVID-19 infection for their own health) on a scale from 1 (not concerned at all) to 10 (very concerned); and (b) risk susceptibility (the perceived likelihood to get COVID-19) on a scale from 1 (very little) to 5 (a lot). Moreover, participants were asked to rate their agreement (from 1, completely disagree, to 5, completely agree) with a series of statements regarding their self-responsibility, self-efficacy in health management, self-efficacy in stress management, the value of partnership in healthcare, trust in science, trust in the National Healthcare System (NHS) and media reliability.

- **Behavioral responses**: involving frequency of information seeking from various different medias (TV, newspapers, social networks, scientific journals etc.) on a scale from 1 (never) to 5 (usually). An average was then calculated, in order to obtain an indicator of how much a certain subject was searching for information regarding the virus. Moreover, a series of dichotomous, yes/no questions were asked regarding changes in consumer habits and, in particular, asking whether they had reduced restaurant meals, ethnic restaurant meals, and the purchase of products coming from "red" zones. They were also asked if they "stockpiled" food and first need products. Finally, they were asked a series of questions surveying whether the buying of different products (i.e., fresh food, frozen food, canned food, products for personal disinfection and care, and products for house disinfection) had diminished/remained the same/increased.

A list of all the items included in the present study has been made available in both English and original language in the Supplementary Information along with the dataset.

## Statistical analysis

**The revised form of PHE scale evaluation.** To evaluate the psychometric properties of the revised PHE-s® scale, a Partial Credit Rasch Model (PCM) was performed to check unidimensionality and the fit of each item at the construct of interest [55]. In the family of Rasch Models, PCM was chosen because the revised PHE-s® items had more than two response options and they showed different patterns of usage [56,57]. It is reasonable to assume that since the thresholds are different for all the items, i.e., each item has its own unique rating scale structure, the PCM appears the most appropriate model. The parameters of the model are estimated by the maximum likelihood method [58]. Then, the Person Separation Index (PSI) was calculated to evaluate the reliability in the Rasch Model. The PSI is an indicator of the quality of measures and refers to the reproducibility of the measured location of the persons. PSI indicates the degree to which study participants can be differentiated into certain groups (range 0–1). Values for PSI superior to 0.8 are acceptable [59,60].

In particular, to check whether the items fitted the expected model, two items fit mean square (MNSQ) statistics (Infit and Outfit) were computed. If the data fit the Rasch model, the fit statistics should be between 0.6 and 1.4 [61]. Analyses of difficulty and step parameters were conducted to guarantee a sufficient ranking of the different categories of response and to respect the monotonic order. The internal consistency of the items of the revised PHE-s® was assessed using Ordinal Alpha via Empirical Copula Index [62] due to the ordinal nature of the items. A reliability index superior to 0.7, 0.8, 0.9 can be interpreted as acceptable, good and excellent, respectively [63].

Finally, a Confirmatory Factor Analysis (CFA) was performed. Goodness-of-fit indexes (i.e. comparative fit index CFI, root mean square residual RMR and root mean square error of approximation RMSEA) were evaluated. A CFI > .90 was considered a good model fit [64], a RMR < .05 was desirable [65], whereas a RMSEA < .08 indicated an acceptable fit [66].

**Attitudinal response towards the COVID-19 health emergency.** To explore differences in the perceived Risk Severity and Risk Susceptibility between different health engagement groups, two factorial ANOVA with Risk Severity and Risk Susceptibility as dependent variables and health engagement and "Coming from red zones" as independent variables were carried out, followed by Tuckey HSD post-hoc tests. Tukey HSD post-hoc test was preferred since it is conservative when there are unequal sample sizes.

To explore the difference in self-responsibility, self-efficacy in health management, self-efficacy in stress management, the value of partnership in healthcare, trust in science, trust in the National Healthcare System (NHS) and perceived media reliability among different health engagement groups, a series of univariate Welch's ANOVA with health engagement as independent variable was carried out followed by Games-Howell post-hoc comparisons. Welch's ANOVA and G-H post-hoc comparisons were preferred over a classic ANOVA approach to provide a more robust method for data analysis [67] since some dependent variables were violating the assumption of homoscedasticity.

**Behavioral responses.** A Welch's ANOVA with Health Engagement as independent variable and Information Seeking as dependent was carried out, followed by Games-Howell post-hoc comparisons to investigate whether people in different health engagement positions show different amounts of media access.

Dichotomous variables were used as dependent variables in a series of multi-variable logistic regressions, with Health Engagement, Risk Susceptibility and Risk Severity as independent

variables Wald forward method was selected to automatically exclude non-significant predictors. Health Engagement was used as a categorical variable and hence dummy coded: Eudaimonic Project was used as the 0, the baseline of comparison, for the other two levels. Dependent variables were coded so that "No" was used as the comparison level for "Yes". Hence, an Odds Ratio > 1 should be interpreted as "more likely to answer yes" and vice-versa.

To assess the association between change in consumer purchase behaviors and different health engagement levels, a series of contingency tables was created. Pearson's chi-square and Fisher's exact tests were also carried out to reject the null hypothesis that data are randomly distributed across health engagement levels. As post-hoc, standardized residuals were inspected: standardized residuals are calculated as the difference between observed and expected counts of a cell divided by an estimate of its standard deviation. Since they are asymptotically normally distributed with a mean of 0 and standard deviation of 1 under the null hypothesis of independence, as a general rule of thumb, cells with an absolute value of standard residuals above 2 can be considered to significantly contribute to the general chi-square value [68]. For stockpiling behavior, groups were way too unbalanced to proceed with a logistic regression (Yes = 5.6%); hence, an approach based on contingency tables was preferred.

## Results

### Sample

Male participants were 473 (48.9%). Mean age was 44 years (SD = 14; range 18–70). For a more detailed description of the study sample, see Table 1.

### Descriptive statistics

There were no missing data in our dataset. For dichotomous and multiple-choice questions, answer frequencies and "I don't know" answers are reported -where provided- in Table 2. However, in the following analyses, "I don't know" was considered as a missing value. For other variables, descriptive statistics are reported in Table 3. Since very few participants resulted being in "Blackout" position, they were grouped together with participants in "Arousal" to facilitate statistical analyses.

### Psychometric proprieties of the PHE-s® revised version

Table 4 shows the results of the Rasch Analysis to test the psychometric properties of the PHE-s® revised version.

The item statistics ranged from .674 to 1.085 for the infit MSQ and from .616 to 1.187 for the outfit MSQ. These values indicate an acceptable fit of the Rasch Model. The distances between subsequent thresholds showed acceptable distinction between the response options and measurement model fit. The PSI for revised PHE-s® was equal to .851. Rasch Model confirmed the unidimensionality of the revised PHE-s® scale and the fit of each item of the scale to the data.

The revised PHE-s® had a quite good internal consistency, since the value of the Ordinal Alpha via Empirical Copula was equal to .788. Each item contributed significantly to the revised PHE-s® scale score. So, the internal consistency of the revised PHE-s® was satisfactory.

CFA showed reasonable goodness of fit indices. The fit indices met the criteria of fit for the hypothesized one-factor structure. All goodness of fit indices (CFI = 0.994, RMR = 0.008, RMSEA = 0.066) suggested that the model is coherent with the data. The analysis of

**Table 1. Demographic profiles of the sample (N = 968).**

| | n | % study sample | % Italian population | | n | % study sample | % Italian population |
|---|---|---|---|---|---|---|---|
| **Gender** | | | | **Having a chronic disease** | | | |
| Male | 473 | 48.9 | 49.3 | Yes | 174 | 18.0 | - |
| Female | 495 | 51.1 | 50.7 | No | 794 | 82.0 | - |
| **Age** | | | | **Geographical area** | | | |
| 18–24 | 99 | 10.1 | 10.0 | North-West | 253 | 26.0 | 26.3 |
| 25–34 | 156 | 16.1 | 16.3 | North-East | 178 | 18.4 | 18.6 |
| 35–44 | 209 | 21.6 | 21.5 | Center | 194 | 20 | 19.7 |
| 45–54 | 215 | 22.2 | 22.7 | South and Islands | 343 | 35.4 | 35.5 |
| 55–59 | 106 | 11.0 | 10.8 | | | | |
| 60–70 | 183 | 19.0 | 18.8 | | | | |
| **Education** | | | | **Coming from "red zones" of the virus** | | | |
| Middle school or lower | 142 | 14.6 | - | Yes | 294 | 30.3 | - |
| High school | 586 | 60.6 | - | No | 674 | 69.7 | - |
| University degree | 240 | 24.8 | - | | | | |
| **Employment** | | | | **Living centre's size** | | | |
| Entrepreneur / freelancer | 119 | 12.3 | 12.4 | Up to 10,000 inhabitants | 313 | 32.3 | 32.1 |
| Manager / official / middle manager | 36 | 3.7 | 3.8 | 10,001/100,000 inhabitants | 430 | 44.4 | 44.0 |
| Employee / teacher / military | 170 | 17.6 | 19.2 | 100,001/500,000 inhabitants | 102 | 10.6 | 10.9 |
| Worker / shop assistant / apprentice | 202 | 20.9 | 21.0 | More than 500,000 inhabitants | 117 | 12.1 | 12.9 |
| Housewife | 146 | 15.1 | 15.0 | Missing | 6 | 0.6 | |
| Student | 54 | 5.5 | 5.3 | | | | |
| Retired | 77 | 7.9 | 7.9 | | | | |
| Unoccupied | 147 | 15.2 | 15.4 | | | | |
| Other | 17 | 1.8 | | | | | |

modification indices did not find any relation between the error covariance of the items. All the standardized to factor loadings ranged from .532 to .820.

## Attitudinal response towards to the COVID-19 health emergency

**Risk severity.** ANOVA results show a significative main effect of Health Engagement on Risk Susceptibility [$F_{(2, 1048)}$ = 185.709; p < .001; $\eta^2_p$ = .262]. No other significant main effect or interaction was found. Tukey post-hoc comparisons show that the Arousal group (M = 8.00; SD = 1.71) was more concerned than either the Adhesion group (M = 5.98; SD = 2.09) or the Eudaimonic Project group (M = 3.51; SD = 2.39) with a significance level of 99.9%. Also, the means difference of Adhesion and Eudaimonic Project groups was found to be statistically significant with p < .001.

**Risk susceptibility.** Results show a significative main effect of Health Engagement on Risk Susceptibility [$F_{(2, 1040)}$ = 150.890; p < .001; $\eta^2_p$ = .225]. No other significant main effect or interaction was found. Tukey post-hoc comparisons revealed that the Arousal group (M = 3.73; SD = .87) perceived themselves as more at risk than either the Adhesion group (M = 2.94; SD = .92) or the Eudaimonic Project group (M = 1.97; SD = .906) with a significance of 99.9%. Also the means difference of Adhesion and Eudaimonic Project groups was found to be statistically significant with p < .001.

**Orientation towards health management and trust in authorities.** ANOVA results show a significant main effect of Health Engagement on Self-Responsibility [$F_{(2, 322.257)}$ = 3.700; p = .026; $\eta^2$ = .009], Self-Efficacy in Health Management [$F_{(2, 339.819)}$ = 57.382; p < .001;

**Table 2. Frequency distribution of items.**

| | n | % | | n | % |
|---|---|---|---|---|---|
| **Health engagement level** | | | **Products from the "red zones"** | | |
| Blackout | 11 | 1.1 | Yes | 498 | 51.1 |
| Arousal | 207 | 21.4 | No | 182 | 18.8 |
| Adherence | 595 | 61.5 | I don't know (missing) | 288 | 29.7 |
| Eudaimonic Project | 155 | 16.0 | | | |
| **Reduced restaurant meals** | | | **Stockpiling** | | |
| Yes | 323 | 33.3 | Yes | 52 | 5.3 |
| No | 645 | 66.7 | No | 916 | 94.7 |
| **Reduced ethnic restaurant meals** | | | **Fresh food** | | |
| Yes | 332 | 34.2 | Diminished | 15 | 1.4 |
| No | 636 | 65.8 | Unchanged | 872 | 90.1 |
| | | | Increased | 76 | 7.9 |
| | | | Not buying (missing) | 5 | .6 |
| **Frozen food** | | | **Personal care** | | |
| Diminished | 13 | 1.2 | Diminished | 10 | 1.0 |
| Unchanged | 867 | 89.6 | Unchanged | 848 | 87.6 |
| Increased | 69 | 7.2 | Increased | 91 | 9.4 |
| Not buying (missing) | 19 | 1.9 | Not buying (missing) | 19 | 2.0 |
| **Canned food** | | | **Personal disinfection** | | |
| Diminished | 17 | 1.7 | Diminished | 9 | .8 |
| Unchanged | 821 | 84.9 | Unchanged | 735 | 76.0 |
| Increased | 98 | 10.1 | Increased | 185 | 19.2 |
| Not buying (missing) | 32 | 3.3 | Not buying (missing) | 39 | 4.1 |
| **House disinfection** | | | | | |
| Diminished | 12 | 1.3 | | | |
| Unchanged | 780 | 80.6 | | | |
| Increased | 142 | 14.7 | | | |
| Not buying (missing) | 34 | 3.4 | | | |

$\eta^2 = .113$], Self-Efficacy in Stress Management [$F_{(2, 355.911)} = 16.497$; $p < .001$; $\eta^2 = .032$], Value of Partnership in Healthcare [$F_{(2, 344.585)} = 9.568$; $p < .001$; $\eta^2 = .022$], Trust in Science [$F_{(2,}$

**Table 3. Descriptive statistics for items.**

| Variable name | Min | Max | Mean | SD | Skewness | Kurtosis |
|---|---|---|---|---|---|---|
| Risk severity | 1 | 10 | 6.04 | 2.48 | -.440 | -.626 |
| Risk susceptibility | 1 | 5 | 2.96 | 1.05 | .054 | -.511 |
| Self-responsibility | 1 | 5 | 3.74 | .920 | -.621 | .418 |
| Information seeking | 1 | 5 | 2.50 | .732 | .520 | -.039 |
| Self-efficacy in health management | 1 | 5 | 3.77 | .719 | -.428 | .920 |
| Self-efficacy in stress management | 1 | 5 | 3.76 | .763 | -.586 | .843 |
| Value of partnership in healthcare | 1 | 5 | 4.06 | .732 | -.610 | .825 |
| Trust in science | 1 | 5 | 4.09 | .874 | -.929 | .949 |
| Trust in the National Health System | 1 | 5 | 3.66 | .934 | -.570 | .275 |
| Media reliability | 1 | 5 | 2.86 | 1.14 | .081 | -.662 |

SD = Standard Deviation.

**Table 4. Partial credit model and item fit statistics.**

| Item | Location | Step 1 | Step 2 | Step 3 | Outfit MSNQ | Infit MSNQ |
|------|----------|--------|--------|--------|-------------|------------|
| Health Engagement 1 | 2.462 | -1.754 | 2.008 | 7.135 | 1.187 | 1.085 |
| Health Engagement 2 | 1.369 | -3.139 | 1.282 | 5.963 | 0.682 | 0.721 |
| Health Engagement 3 | 0.547 | -2.785 | 1.172 | 3.254 | 0.616 | 0.674 |
| Health Engagement 4 | 1.075 | -2.186 | 1.081 | 4.331 | 0.773 | 0.728 |
| Health Engagement 5 | 0.991 | -2.531 | -0.086 | 5.591 | 0.642 | 0.699 |

$_{335.022}$) = 8.158; p = .001; $\eta^2$ = .018], Trust in NHS [$F_{(2, 337.641)}$ = 9.575; p < .001; $\eta^2$ = .021] and Media Reliability [$F_{(2, 344.288)}$ = 28.664; p < .001; $\eta^2$ = .060]. Results of Games-Howell comparisons are reported in Table 5.

## Behavioral responses

**Information seeking.** ANOVA results show a significant main effect of Health Engagement on Information Seeking [$F_{(2, 334.095)}$ = 29.344; p < .001; $\eta^2$ = .064]. G-H comparisons showed that the amount of Information Seeking differed significantly among all the different levels: in particular, results showed that people in Arousal search significantly more information (M = 2.79; SD = .74) than people in either Adhesion (M = 2.47; SD = .68) or Eudaimonic project (M = 2.20; SD = .77). The comparison between Adhesion and Eudaimonic project was significantly different as well.

**Consumer habits and purchasing behaviors.** Results of the logistic regressions are reported in Table 6. In particular, results show that higher levels of Risk Severity and Risk Susceptibility are associated with a higher probability of having reduced meals outside in both generic and ethnic restaurants. Perceived Risk Severity is also a predictor of the willingness to buy products coming from "red zones" (higher perceived Severity increases the probability of not being willing to buy). Results also show that Health Engagement (HE) levels predict having reduced meals outside (lower levels of engagement have a higher probability) and of being willing to buy "red zone" products (lower engagement, lower probability).

Results of contingency tables are reported in Table 7. Pearson's chi-squared analysis and the inspection of standardized residuals show that different levels of Health Engagement are associated with different consumer behaviors: in particular, our results show that lower levels

**Table 5. Results of Games-Howell comparisons.**

| Dependent variables | Engagement Level Comparison | | |
|---------------------|-----------------|----------------|------------------|
| | Arousal-Adhesion | Arousal-Eudaimonic | Adhesion-Eudaimonic |
| Self-responsibility | -.162 (.073) | -.274 (.110)* | -.112 (.095) |
| Self-efficacy in health management | -.326 (.057)*** | -.791 (.074)*** | -.465 (.060)*** |
| Self-efficacy in stress management | -.122 (.059) | -.434 (.077)*** | -.312 (.066)*** |
| Value of partnership in healthcare | -.205 (.062)** | -.335 (.078)*** | -.130 (.062) |
| Trust in science | -.218 (.071)** | -.378 (.099)** | -.160 (.081) |
| Trust in the National Health System | -.245 (.072)** | -.425 (.104)** | -.181 (.091) |
| Media | -.352 (.084)*** | -.911 (.120)*** | -.559 (.107)*** |

values in cells are differences in means. Standard errors are reported in brackets. Significance in marked with asterisks

(* sig. at p < .05

** sig. at p < .01

***sig at p < .001).

**Table 6. Results of logistic regressions.**

| Behaviors | Variables | B | S.E. | Wald | P | Odds Ratio |
|---|---|---|---|---|---|---|
| Reduced restaurant meals *Nagelkerke's R² = .232 Correctly predicted: 72.0% Chi-square = 174.63 (d.f. = 4), p < .001* | Health Engagement | | | 15.176 | .001 | |
| | Health Engagement (Arousal) | .823 | .321 | 6.579 | .010 | 2.277 |
| | Health Engagement (Adhesion) | .110 | .275 | .161 | n.s. | |
| | Risk Severity | .244 | .047 | 27.441 | < .001 | 1.276 |
| | Risk Susceptibility | .285 | .097 | 8.526 | .004 | 1.329 |
| Reduced ethnic restaurant meals *Nagelkerke's R² = .170 Correctly predicted: 70.1% Chi-square = 124.92 (d.f. = 4), p < .001* | Health Engagement | | | 11.449 | .003 | |
| | Health Engagement (Arousal) | .799 | .309 | 6.703 | .010 | 2.223 |
| | Health Engagement (Adhesion) | .210 | .260 | .651 | n.s. | |
| | Risk Severity | .195 | .044 | 19.638 | < .001 | 1.216 |
| | Risk Susceptibility | .221 | .094 | 5.029 | .025 | 1.235 |
| Products from the "red zones" *Nagelkerke's R² = .146 Correctly predicted: 75.5% Chi-square = 70.954 (d.f. = 3), p < .001* | Health Engagement | | | 12.032 | .002 | |
| | Health Engagement (Arousal) | -1.313 | .408 | 10.372 | .001 | .269 |
| | Health Engagement (Adhesion) | -.681 | .349 | 3.808 | .051 | .506 |
| | Risk Severity | -.190 | .047 | 16.365 | < .001 | .827 |

df = degrees of freedom; HE = Health Engagement; S.E. = Standard Error; P = p-value.

of engagement are more frequently associated with stockpiling, and with an increased consumption of fresh, canned and frozen food, and with products for disinfection when compared with average and high levels of engagement.

## Discussion

By the end of February 2020, the diffusion of the COVID-19 epidemics in northern Italy had forced health authorities to embrace restrictive preventive measures that impacted Italian citizens' daily habits and consumption behaviors. Enforcing compliance with such measures was crucial at that time in order to enhance their effectiveness and to sustain the sustainability of the healthcare system. However, this sudden change caused huge reactions by Italian citizens: many of them experienced panic and enacted maladaptive behaviors (for example the migration from north to south Italy immediately after the Lombardy region lockdown, which was initially considered a "red zone"; also, food stockpiling happened soon after the first cases of Covid-19 disease came out, which created negative consequences for the food chain organization). In this scenario, the Italian citizens' reactions to the COVID-19 emergency measures, from the scientific perspective, is an interesting and unique platform to demonstrate the value of making citizens engaged as actual partners of the healthcare system to safeguard both individual and collective health. Therefore, we consider the current COVID-19 outbreak in Italy as a valuable "testing ground" for consumer education initiatives aimed at sustaining their health engagement and compliance with the prescribed behavioral changes. Existing research has focused on demographic and immutable and subjective factors that influence how people are likely to behave in a pandemic [69–71]. Furthermore, previous research on responses to

**Table 7. Results of contingency tables.**

| Variables | Answers | Cell | Health Engagement level | | | Row Total |
|---|---|---|---|---|---|---|
| | | | Arousal | Adhesion | Eudaimonic project | |
| Stockpiling *Chi-square = 23.659(df = 2), p < .001 Fisher = 20.989, p < .001* | No | Observed | 192 | 570 | 153 | 915 |
| | | Expected | 205.5 | 562.6 | 146.8 | |
| | | Std res. | -.9 | .3 | .5 | |
| | Yes | Observed | 25 | 24 | 2 | 51 |
| | | Expected | 11.5 | 31.4 | 8.2 | |
| | | Std res. | **4.0** | -1.3 | **-2.2** | |
| | | CT | 217 | 594 | 155 | |
| Fresh food *Chi-square = 23.562(df = 4), p < .001 Fisher = 20.419, p < .001* | Diminished | Observed | 3 | 10 | 1 | 14 |
| | | Expected | 3.1 | 8.6 | 2.2 | |
| | | Std res. | -.1 | .5 | -.8 | |
| | Unchanged | Observed | 179 | 547 | 145 | 871 |
| | | Expected | 195.1 | 538.0 | 137.9 | |
| | | Std res. | -1.2 | .4 | -.6 | |
| | Increased | Observed | 33 | 36 | 6 | 75 |
| | | Expected | 16.8 | 46.3 | 11.9 | |
| | | Std res. | **4.0** | -1.5 | -1.7 | |
| | | CT | 215 | 593 | 152 | |
| Canned food *Chi-square = 44.238(df = 4), p < .001 Fisher = 39.352, p < .001* | Diminished | Observed | 4 | 7 | 5 | 16 |
| | | Expected | 3.6 | 9.9 | 2.5 | |
| | | Std res. | .2 | -.9 | 1.6 | |
| | Unchanged | Observed | 159 | 526 | 136 | 821 |
| | | Expected | 183.7 | 508.1 | 129.2 | |
| | | Std res. | -1.8 | .8 | .6 | |
| | Increased | Observed | 46 | 45 | 6 | 95 |
| | | Expected | 21.7 | 60.0 | 15.3 | |
| | | Std res. | **5.2** | -1.9 | **-2.4** | |
| | | CT | 212 | 580 | 146 | |
| Frozen food *Chi-square = 41.970(df = 4), p < .001 Fisher = 36.015, p < .001* | Diminished | Observed | 4 | 6 | 2 | 12 |
| | | Expected | 2.7 | 7.4 | 1.9 | |
| | | Std res. | .8 | -.5 | .1 | |
| | Unchanged | Observed | 173 | 549 | 145 | 867 |
| | | Expected | 195.5 | 533.5 | 138.0 | |
| | | Std res. | -1.6 | .7 | .6 | |
| | Increased | Observed | 37 | 29 | 4 | 70 |
| | | Expected | 15.8 | 43.1 | 11.1 | |
| | | Std res. | **5.3** | **-2.1** | **-2.1** | |
| | | CT | 214 | 584 | 151 | |
| Personal disinfection *Chi-square = 61.148(df = 4), p < .001 Fisher = 57.087, p < .001* | Diminished | Observed | 3 | 3 | 2 | 8 |
| | | Expected | 1.8 | 4.9 | 1.3 | |
| | | Std res. | .9 | -.9 | .7 | |
| | Unchanged | Observed | 127 | 477 | 131 | 735 |
| | | Expected | 166.1 | 452.6 | 116.3 | |
| | | Std res. | **-3.0** | 1.1 | 1.4 | |
| | Increased | Observed | 80 | 92 | 14 | 186 |
| | | Expected | 42 | 114.5 | 29.4 | |
| | | Std res. | **5.9** | **-2.1** | **-2.8** | |
| | | CT | 210 | 572 | 147 | |

(*Continued*)

**Table 7.** (Continued)

| Variables | Answers | Cell | Health Engagement level | | | Row Total |
|---|---|---|---|---|---|---|
| | | | Arousal | Adhesion | Eudaimonic project | |
| Home disinfection *Chi-square = 73.370(df = 4), p < .001 Fisher = 64.274, p < .001* | Diminished | Observed | 4 | 7 | 2 | 13 |
| | | Expected | 2.9 | 8.0 | 2.0 | |
| | | Std res. | .6 | -.4 | .0 | |
| | Unchanged | Observed | 137 | 509 | 134 | 780 |
| | | Expected | 176.9 | 480.5.6 | 122.6 | |
| | | Std res. | **-3.1** | 1.4 | 1.0 | |
| | Increased | Observed | 71 | 60 | 11 | 142 |
| | | Expected | 32.3 | 87.5 | 22.3 | |
| | | Std res. | **6.8** | **-2.9** | **-2.4** | |
| | | CT | 214 | 576 | 144 | |
| Personal care *Chi-square = 54.049(df = 4), p < .001 Fisher = 46.845, p < .001* | Diminished | Observed | 3 | 5 | 2 | 10 |
| | | Expected | 2.3 | 6.2 | 1.5 | |
| | | Std res. | .5 | -.5 | .4 | |
| | Unchanged | Observed | 164 | 544 | 139 | 847 |
| | | Expected | 192.1 | 524.5 | 130.4 | |
| | | Std res. | **-2.0** | .9 | .7 | |
| | Increased | Observed | 48 | 38 | 5 | 91 |
| | | Expected | 20.6 | 56.3 | 14.0 | |
| | | Std res. | **6.0** | **-2.4** | **-2.4** | |
| | | CT | 215 | 587 | 146 | |

CT = Column Total; Std res = standard residues¸ df = degrees of freedom. Cells with an absolute value of std. res >2 are marked in bold.

pandemics has been largely a-theoretical [11]. Therefore, all these studies provide valuable insights into how different segments of the population are likely to respond, but do not tell us why they respond in this way. The current study adopted the theoretical lens of the Patient Health Engagement Model (PHE) to explain–from a psychosocial perspective—people's responses through the first wave of the COVID-19 pandemic in Italy. This theory states that individuals are more or less likely to change their behaviors according to their own subjective perceptions about the role (more or less active) they might play in their health and care [46].

The Patient Health Engagement Model (PHE) provides a potentially useful framework for understanding how people will respond to health threats such as pandemics and related pre-scribed preventive measures imposed by healthcare authorities. The PHE model proposes that people's adaptive behavioral and emotional responses to protect themselves from a health threat are influenced by their level of health engagement–that is a progressive reframing of individuals' own roles within the healthcare system (i.e. from passive users of services to active partners of the healthcare system) [46]. In this study, we employed and evaluated the psycho-metric properties of a revised version of the PHE-s® to measure citizens' health engagement. This revised version showed good psychometric properties for our representative sample.

According to the study results, Italian citizens seems to be more concerned about the health emergency than not, even though not extremely worried (on a scale from 1 to 10, the average is around 6) and not feeling exceedingly at risk of being infected (the 5-point Likert shows a normaloid distribution with mean around the central point), confirming previous studies in other similar settings [9,72,73]. Nevertheless, it is important to notice how different health engagement profiles are associated with different levels of both perceived risk severity and

susceptibility: indeed, less engaged people (rated as in "Blackout" and "Arousal") show significantly higher levels of perceived susceptibility to and perceived risk of the infection when compared with highly engaged ones (rated as in "Adhesion" and "Eudaimonic Project"), regardless of the geographical area of origin ("red zone" or not), which surprisingly wasn't found to be associated with different levels of susceptibility and severity. This seems to support that people differ in their ability to psychologically master their worries related to the COVID-19 epidemic, and this explains the consequent more or less adherence to the change in behaviors imposed by the health authorities. This interpretation is confirmed also by the fact that people with different levels of health engagement show different attitudinal responses to the emergency: in particular, when compared to people with higher levels of health engagement, less engaged people are less trusting of scientific and healthcare authorities, they feel less self-effective in managing their own health—both in normal conditions and under stress—and are less prone to cooperate with healthcare professionals [74]. These results confirm previous studies on Influenza A (H1N1), which demonstrated that if perceived severity and susceptibility are high but response and self-efficacy are low, maladaptive responses (e.g. denying the existence of a threat) are likely to ensue [75]. The perceived self-efficacy in health management and a sentiment of mistrust towards authorities may actually help in understanding why a less engaged person feels more concerned and worried about the new COVID-19 emergency: they seek more information, potentially exposing themselves to fake or over-hyped news, since they are also prone to feel that news regarding the emergency is reliable; nevertheless, they mistrust scientific research and the capacity of the NHS to cope with the pandemic and feel less capable of taking care of themselves. Furthermore, low levels of health engagement may demonstrate that people do not consider themselves ready to be active partners of the healthcare systems, being more focused on their own health interests and need and not inclined to collaborate and trust the healthcare system to achieve a common public health goal.

The health engagement construct also seems to be a predictor of behavioral responses to the emergency. Generally speaking, a substantial part of our sample reported a change in their habits: one out of three Italian citizens reported having fewer meals outside and/or meals in ethnic restaurants, while 20% declared that they would not buy products coming from "red zones." Indeed, while risk severity and risk susceptibility are clearly strong predictors, logistic models show that people with lower levels of health engagement are more than twice as likely as people with higher level of health engagement to have reduced either meals or ethnic meals outside their home. It's important to notice that data have been collected at a moment when the emergency was still away from its peak and guidelines were not forbidding people from moving freely or from having meals in restaurants. These results could be interpreted as in line with previous studies underlining that when unknown diseases are thought to be lethal, people are inclined to blame the outbreaks on someone, or some group of people, who live outside of their own social sphere, as a mechanism to cope with fear and risk perception [76]. In this research, it appears clear that this form of "moral panic" [77] had a halo effect also on products and restaurants people naively thought were guilty in the Covid-19 disease spread, or that were related to the "infected zone." Such lay interpretation of disease transmission, together with the difficulty of finding reliable information in a first phase of health emergency, has an impact on people's habits and consumptions, and clear consequences for the local enterprises' economy. A similar case occurred with the H5N1 Avian Influenza on food consumption, when the poultry industry suffered severe losses due to a sort of "halo effect" in consumer perception of risk, even after the emergency was over [78,79].

Despite these results, with respect to buying behaviours, our data show that generally, most people didn't actually change their habits, in line with other studies [80]: most people didn't stockpile goods or increase the purchase of the goods we considered in our survey.

Nevertheless, crosstabs show that amongst those who stockpiled goods and increased the purchase of food (fresh, frozen or canned) and disinfection products (in particular regarding home disinfection), there is a significantly higher presence of lower engaged consumers. This evidence is in line with other studies [81–83] that showed how personal reaction to the critical event can feed behavioral changes, with many people making significant changes in their consumption behaviors like anticipating the purchasing of goods [84,85]. As food consumption is recognized as a primary need for individuals, it is strongly influenced by the subjective interpretation of risk and the possible scarcity [86]. For this reason, these results appear interesting in giving a sense of how people orient their food purchase in the case of emergency in relation to their engagement level [87,88]. Furthermore, it appears evident that people with a low level of health engagement, not being psychologically ready to consider the social and public health consequences of their conduct, appear more focused on their own health interests and less keen to rely on health authorities' guidelines to orient their behaviors [3,89]. For instance, the behavior of stockpiling goods carried out by the less engaged Italian citizens had a negative organizational impact on food supplies, which further compromised the delicate situation of the Italian population. Furthermore, the overcrowding at superstores in the situation of the COVID-19 epidemic was highly counterproductive and contributed to spreading the risk of contagion.

## Limitations and future studies

The study measured a specific population's views at a specific point in time; their beliefs and attitudes reflect the information available at the time and therefore are not stable. Second, results were self-reported and data were collected through a broader continuative online-based survey: measurement errors, unreliable answers and social desirability bias may have partially altered results, as is generally the case in these kinds of studies. Finally, this study relies on several *ad hoc* items, specifically developed for our research questions but not validated; regardless of our effort to make them clear and non-ambiguous, it is still possible that some participants may have misinterpreted them. Two items in particular (those regarding reduced meals in restaurants and ethnic restaurants) may raise some concerns as participants were not given the option to answer that they never go to (ethnic) restaurants.

Future research should test the Patient Health Engagement Model as a predictor of particular preventive behaviors in different socio-cultural contexts. This model is indeed relatively young and current evidence about its applicability has been carried out mainly by the same research group who developed it. Further studies are needed to consolidate it and to confirm the reliability of the results on the larger Italian population. In addition, it is important to carry out further behavioral research where actual behavior can be measured, not only self-reported.

## Practical implications

This study has provided evidence about the role of health engagement as a determinant of citizens' behavioral change, which is key for controlling the spread of pandemic disease, and has described a conceptual framework–i.e. the Patient Health Engagement Model—in which to better understand these behaviors [90]. In sum, the study shows that health engagement levels are predictive of different responses, both affective and behavioral: playing an active role in health management is associated with a higher chance of performing specific behaviors. In particular, the psychological readiness to assume a proactive role in their own health prevention depends on the individuals' tendency to be more or less able to comply with health authorities' prescriptions and to perceive themselves as mainly responsible for their own health and the health of their community. Furthermore, the psychological readiness to engage in

health is a crucial factor for explaining the different way in which individuals can cope with their worries about a health emergency. The findings suggest that intervention studies should focus on particular groups and on raising their levels of engagement to increase the effectiveness of educational initiatives designed to promote preventive behaviors. Communication strategies should maximize their impact by targeting messages according to the health engagement levels of citizens. For instance, in order to improve the levels of engagement of citizens in a "psychological blackout," reassuring messages aimed at sustaining the emotional elaboration of the emergency and related worries would be particularly needed. For those citizens, psychological counselling and positive emotions facilitated by a social campaign also are suggested. To enhance the motivation to stay engaged, citizens in a situation of "psychological adherence" would need positive stories of other persons who succeeded in adhering to the prescribed containment measures. For instance, video testimony of peers able to describe how they successfully coped with the emergency, sharing tips and advice. Finally, people in the position of "Eudaimonic Project," who were able to develop a new sense of normality despite the serious emergency, can be involved in peer-to-peer communication initiatives, becoming advocates for the correct engagement in adhering to the prescribed measures to face the COVID-19 epidemic. Furthermore, this target group could be further engaged in an open and accountable debate with healthcare authorities to better understand the rationale of some decisions about containment measures and to contribute raising their voice to orient them. Furthermore, fostering the psychological readiness to get engaged in health prevention appears to be a crucial goal for educational and communication initiatives in the event of a health emergency. Carrying out this work now will be invaluable in preparing for this and future pandemics. Listening to consumers' concerns and expectations in an emergency situation is the base for building a collaborative space where health authorities and civic communities can all contribute to the best management of the situation. Measuring the levels of health engagement of citizens may be considered as a vital parameter for healthcare authorities in order to best orient educational initiatives and supports able to sustain citizens' adherence to the preventative measures.

## Supporting information

**S1 Appendix. Survey coronavirus Eng.** Study survey–English version,
(DOCX)

**S2 Appendix. Survey coronavirus ITA.** Study survey–Italian version.
(DOCX)

**S3 Appendix.**
(PDF)

**S1 File. Coronavirus Eng.** Study original dataset.
(SAV)

## Author Contributions

**Conceptualization:** Guendalina Graffigna.

**Data curation:** Lorenzo Palamenghi.

**Formal analysis:** Lorenzo Palamenghi, Greta Castellini, Andrea Bonanomi.

**Methodology:** Guendalina Graffigna, Serena Barello, Mariarosaria Savarese, Lorenzo Palamenghi, Greta Castellini, Andrea Bonanomi, Edoardo Lozza.

**Project administration:** Guendalina Graffigna.

**Supervision:** Guendalina Graffigna.

**Writing – original draft:** Serena Barello, Mariarosaria Savarese.

**Writing – review & editing:** Guendalina Graffigna, Serena Barello, Mariarosaria Savarese, Lorenzo Palamenghi, Greta Castellini, Andrea Bonanomi, Edoardo Lozza.

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
