## [Decision Letter · Decision Letter 0]

20 May 2020

PONE-D-20-11202

Measuring Italian Citizens’ Engagement in the First Wave of the COVID-19 Pandemic Containment Measures: A Cross-sectional Study

PLOS ONE

Dear Dr. Barello,

Thank you for submitting your manuscript to PLOS ONE. After careful consideration, we feel that it has merit but does not fully meet PLOS ONE’s publication criteria as it currently stands. Therefore, we invite you to submit a revised version of the manuscript that addresses the points raised during the review process.

We would appreciate receiving your revised manuscript by Jul 04 2020 11:59PM. To enhance the reproducibility of your results, we recommend that if applicable you deposit your laboratory protocols in protocols.io, where a protocol can be assigned its own identifier (DOI) such that it can be cited independently in the future. For instructions see: http://journals.plos.org/plosone/s/submission-guidelines#loc-laboratory-protocols

We look forward to receiving your revised manuscript.

Kind regards,

Wen-Jun Tu

Academic Editor

PLOS ONE

Journal Requirements:

2. Please provide additional details regarding participant consent.

In the ethics statement in the Methods and online submission information, please ensure that you have specified (a) whether consent was informed and (b) what type you obtained (for instance, written or verbal).

If your study included minors, state whether you obtained consent from parents or guardians.

If the need for consent was waived by the ethics committee, please include this information.

3. Please include additional information regarding the survey or questionnaire used in the study and ensure that you have provided sufficient details that others could replicate the analyses.

For instance, if you developed a questionnaire as part of this study and it is not under a copyright more restrictive than CC-BY, please include a copy, in both the original language and English, as Supporting Information. Moreover, please include more details on how the questionnaire was pre-tested, and whether it was validated.

5. Thank you for stating the following in the Declarations Section of your manuscript:

'Funding

This study was conducted was conducted within the CRAFT project, funded by Fondazione Cariplo & Regione Lombardia'

'The authors received no specific funding for this work'

6. Your ethics statement must appear in the Methods section of your manuscript. If your ethics statement is written in any section besides the Methods, please move it to the Methods section and delete it from any other section. Please also ensure that your ethics statement is included in your manuscript, as the ethics section of your online submission will not be published alongside your manuscript.

Reviewers' comments:

Reviewer's Responses to Questions

**Comments to the Author**

1. Is the manuscript technically sound, and do the data support the conclusions?

Reviewer #1: Partly

2. Has the statistical analysis been performed appropriately and rigorously? 

Reviewer #1: No

3. Have the authors made all data underlying the findings in their manuscript fully available?

Reviewer #1: No

4. Is the manuscript presented in an intelligible fashion and written in standard English?

Reviewer #1: No

5. Review Comments to the Author

Reviewer #1: The manuscript by Barello et al presents the results of online survey administered between February 28th and March 4th 2020 among 1000 persons in Italy. It has several interesting and potentially important findings, however, the manuscript should be substantially revised to overcome numerous pitfalls listed below.

1. The manuscript requires correction of English, since sometimes there are expressions which hard to understand. The manuscript should be checked for the vocabulary, and some non-existing words like “iperactive” should be corrected.

2. In the “Abstract” the description of statistical methods “To investigate the relationship between Health Engagement and these variables, a series of ANOVAs, Logistic regressions and crosstabs have been carried out” should be changed to avoid plural form of analyses, and clarify what does the “crosstabs” mean.

3. The “Abstract” should include a brief characteristic of the “Health Engagement (PHE-S)” metric.

4. The authors cite their own previously published papers to support the patient health engagement importance, and in general it is normal if not excessive. However, the “patient health engagement” search in PubMed returns over 30,000 results, and more equilibrated balance between self-citation and citation of other research groups would be much better.

5. The only two references supporting the detailed explanations of the “Patient Health Engagement model” at line 116 present the papers published by the authors. If this model is unique and has not been applied by others, this should be stated explicitly. The lack of validating studies for this model from independent research groups (if this is a case) should also be indicated.

6. Moreover, the description of the model itself at lines 116-140 is not very clear and should be revised. The consequences of four model positions should be provided. The description of the first “blackout” phase (in the authors’ terminology) as “psychologically frozen and behaviorally paralyzed” is not very compliant with the current epidemic, since many persons continued to follow their routine life style soon after the initial epidemic. The overall details provided for the model is very scarce.

7. The statement at line 201 “For this study purposes the PHE- s® was slightly revised in order to adapt the items formulation to the specific contest of the health emergency” indicates the modification of the original model. But no details provided about this “slightly revised” model.

8. In the “Methods” it is not clear how “using random digit dialing” the participants were selected for the online survey.

9. The authors stated that almost 3% of involved survey participants were excluded because they indicated during the survey wrong age or sex compared to initially registered in the commercial panel they used. But if the discrepancy for sex or age is as high as 3%, how we could be confident that the participants provided the meaningful answers to other questions and not just click over answers to receive a financial incentives for completing the survey?

10. While the authors stated the sample was representative, they did not provide comparison to the Italian adult population data, and the representativeness of the sample remains questionable.

11. At line 182 the authors stated that “Ethical approval was obtained from the institutional review board at Catholic University of Milan (IRB#2019-12)”, that seemingly indicate the IRB occurred in December 2019, when nobody known about COVID-19 epidemic. Please could you address this issue, and provide the Ethical approval in the Supplement?

12. The idea to study the trust to authorities is very interesting, but the questions used seem questionable. Particularly, the statement “I fully trust the National Healthcare System” rated by on a scale ranging between 1 (completely disagree) to 5 (completely agree) is very non-specific, and could reflect not the trust to authorities but believes about financial stability of the NHS. This is a strong limitation, since it is among the central topics of the manuscript, and requires a caution reflection in the “Discussion”.

13. Moreover, the question about media reliability (lines 228-231) is not related at all to the trust to authorities, and thus the whole analysis on measurement of trust to authorities become very questionable.

14. The question “have you reduced meals in ethnic restaurants?” at line 244 has no sense for persons who not visiting ethnic restaurants.

15. The description of questions regarding changes in purchasing behaviors at lines 250-260 is limited and does not contain the possible answer.

16. As a general consideration, instead of describing all questions in the “Study measures” section, the authors could provide the complete questionnaire used during survey in the Supplement, and describe in the “Study measures” more precisely the meaning of the questions used.

17. The presentation of the “Results” in Table 1, 3 and 4, 7 is not compliant for publication but resembles more the output from statistical software.

18. Table 1 contains the description of the classification of participants according to the “Chronic patient” category, but any details about its definition are absent in the “Methods”.

19. The “Partial Credit Rasch Model” described in “Results” should be described in the “Methods”, and of course it should be supported by the relevant references to the literature (that is insufficient in the manuscript now).

20. Tables should not include abbreviations in the title, or should include footer with their explanations.

21. The authors introduce some concepts “on the fly” in the “Results” without any explanations in the “Methods”, like “Person Separation Index”.

22. The calculation of the “Health Engagement Level“ is not clear, and not described in the manuscript. The authors should provide the exact explanations on which questions they used for defining one of the four possible levels. Without knowing how it was calculated, all subsequent discussion about the role of Health Engagement (that is a central idea of the manuscript) seems not having substantial ground.

23. The same concerns also all metrics mentioned in the Table 4. How the authors calculated “Risk severity”.

24. The description of differences between groups provided at lines 333-336 is repetitive. The same concerns lines 343-346.

25. Many lines that authors put in “Results” to describe statistical methods should be used in “Methods”.

26. Table 6 should not use plural for “logistic regression”, but most importantly should clearly define whether it was uni- or multivariable LR.

27. In the “Discussion” the authors stated that “many of Italian citizens experienced panic and enacted maladaptive behaviors”, without any examples that should be provided.

28. The discussion is rather weak and includes rather few references to studies measuring impact of previous epidemics in other countries. This should be corrected.

29. “Limitations” should include issues pointed above.

30. In the “Practical implications” the authors propose, for example to provide during the “ “psychological blackout”, reassuring messages, aimed at sustaining the emotional

elaboration of the emergency and related worries would be particularly needed”. But they did not clarify what messages they exactly mean, and did not refer previously to any literature suggesting the content of such “reassuring messages”. The same concerns also other proposed by the authors stages of “Patient Health Engagement”. Thus, the practical implications remain rather vague.

6. PLOS authors have the option to publish the peer review history of their article (what does this mean?). If published, this will include your full peer review and any attached files.

Reviewer #1: No

---

## [Author Response · Author response to Decision Letter 0]

2 Jul 2020

Comments from the editors and reviewer

We are very grateful to have been given the opportunity to revise our manuscript for Plos One.

We thank the referee and editor for their comments to strengthen the presentation of our work. We have modified the text to respond to all the issues and have elaborated on the changes below.

Editor Comments

Thanks. We revised accordingly.

2. Please provide additional details regarding participant consent. In the ethics statement in the Methods and online submission information, please ensure that you have specified (a) whether consent was informed and (b) what type you obtained (for instance, written or verbal). If your study included minors, state whether you obtained consent from parents or guardians. If the need for consent was waived by the ethics committee, please include this information.

We thank the Editors for the suggestions. We have provided additional information regarding the participant consent in the Ethic statement in the methods section.

3. Please include additional information regarding the survey or questionnaire used in the study and ensure that you have provided sufficient details that others could replicate the analyses.

For instance, if you developed a questionnaire as part of this study and it is not under a copyright more restrictive than CC-BY, please include a copy, in both the original language and English, as Supporting Information. Moreover, please include more details on how the questionnaire was pre-tested, and whether it was validated.

Thank you, we have provided a copy of both the original survey and an English translation as Supplementary Information.

Raw anonymous data have been uploaded as supplementary information in a .sav format (commonly used in psychology and social sciences, usable with software such as SPSS, R, Jasp).

We have made our anonymized dataset available and provided it as as Supporting information materials

5. Thank you for stating the following in the Declarations Section of your manuscript:

'Funding This study was conducted was conducted within the CRAFT project, funded by Fondazione Cariplo & Regione Lombardia'

We note that you have provided funding information that is not currently declared in your Funding Statement. However, funding information should not appear in the Acknowledgments section or other areas of your manuscript. We will only publish funding information present in the Funding Statement section of the online submission form. Please remove any funding-related text from the manuscript and let us know how you would like to update your Funding Statement. Currently, your Funding Statement reads as follows: 'The authors received no specific funding for this work' Please include your amended statements within your cover letter; we will change the online submission form on your behalf.

Dear Editor, thank you for the suggestion. We have deleted the statement from the main text and we kindly ask you to update it in the online form. Please, add following the statement. 

Funding: This study was conducted was conducted within the CRAFT project, funded by Fondazione Cariplo & Regione Lombardia.

6. Your ethics statement must appear in the Methods section of your manuscript. If your ethics statement is written in any section besides the Methods, please move it to the Methods section and delete it from any other section. Please also ensure that your ethics statement is included in your manuscript, as the ethics section of your online submission will not be published alongside your manuscript.

Thanks for the suggestion. We moved the ethic statement in the Methods section as suggested. 

Reviewer’s comments 

Reviewer #1: The manuscript by Barello et al presents the results of online survey administered between February 28th and March 4th 2020 among 1000 persons in Italy. It has several interesting and potentially important findings, however, the manuscript should be substantially revised to overcome numerous pitfalls listed below.

We thank the Reviewer for all the suggestions, which we have addressed and explained in the following points. The suggestions allowed us to better clarify some crucial points of the article and to revise it. We hope that this could improve our contribution to the study of Covid-19 psychological fallouts on the Italian population. We remain at disposition if further clarifications are needed. 

1. The manuscript requires correction of English, since sometimes there are expressions which hard to understand. The manuscript should be checked for the vocabulary, and some non-existing words like “iperactive” should be corrected.

After Reviewer’s suggestions, we have the article revised by a professional native English reviewer. Certification is available upon requests.

2. In the “Abstract” the description of statistical methods “To investigate the relationship between Health Engagement and these variables, a series of ANOVAs, Logistic regressions and crosstabs have been carried out” should be changed to avoid plural form of analyses, and clarify what does the “crosstabs” mean.

We thank the Reviewer for the suggestions that allowed us to better clarify the abstract description. We have revised it avoiding plural forms in the analyses description and we have substitute “crosstabs” with “contingency table”, in order to better specify the nature of this analysis. 

3. The “Abstract” should include a brief characteristic of the “Health Engagement (PHE-S)” metric.

Thanks to the Reviewer’s suggestions, we have added a brief description of the PHE-S metrics in the Abstract section. 

4. The authors cite their own previously published papers to support the patient health engagement importance, and in general it is normal if not excessive. However, the “patient health engagement” search in PubMed returns over 30,000 results, and more equilibrated balance between self-citation and citation of other research groups would be much better.

We have also increased the number of publications by other authors suggesting the relevance and explicative power of patient engagement on a series of clinical and behavioural outcomes to better support our background and research questions. 

5. The only two references supporting the detailed explanations of the “Patient Health Engagement model” at line 116 present the papers published by the authors. If this model is unique and has not been applied by others, this should be stated explicitly. The lack of validating studies for this model from independent research groups (if this is a case) should also be indicated.

Dear Reviewer, thank you for the suggestions. The Patient Health Engagement Model is a relatively new model developed by the authors in 2017. Until now, the model has been validated in China, Spain, Korea and is under cross-cultural validation in other languages by independent authors, confirming its metric characteristics and explicative value (references added in the main text). Moreover, it has been used by some research groups in collaboration with the authors (references added in the main text). We are also aware about the newness of this model and further research is needed in order to strengthen these first results. So, together with the addition of new references, we have also addresses this aspect in the limitation of the study. 

6. Moreover, the description of the model itself at lines 116-140 is not very clear and should be revised. The consequences of four model positions should be provided. The description of the first “blackout” phase (in the authors’ terminology) as “psychologically frozen and behaviorally paralyzed” is not very compliant with the current epidemic, since many persons continued to follow their routine life style soon after the initial epidemic. The overall details provided for the model is very scarce.

Thank to the Reviewer for the comments. We have broadened the description of PHE model and given a theoretical revision aimed at better connect the model with the context of application of COVID-19 disease emergency. We hope that this can better clarify the doubts raised. We remain at disposition if further clarifications are needed. 

We also think the overall description has taken advantage from the English revision.

7. The statement at line 201 “For this study purposes the PHE- s® was slightly revised in order to adapt the items formulation to the specific contest of the health emergency” indicates the modification of the original model. But no details provided about this “slightly revised” model.

We have added a brief description in the same section of what we have adapted in order to answer to this Reviewer’s comment. Thank you. Briefly, the scale was revised in the incipit and in some items (reported in detail in the main text) to adapt to the specific situation. 

8. In the “Methods” it is not clear how “using random digit dialing” the participants were selected for the online survey.

We thank the reviewer for the comment. We clarified in the methods section the sampling procedures. 

9. The authors stated that almost 3% of involved survey participants were excluded because they indicated during the survey wrong age or sex compared to initially registered in the commercial panel they used. But if the discrepancy for sex or age is as high as 3%, how we could be confident that the participants provided the meaningful answers to other questions and not just click over answers to receive a financial incentives for completing the survey?

Thank you for pointing this out. Reliability is an issue with every online-based data collection, as the researcher has (generally) no means to ensure participants’ compliance and data quality. In this particular case, we decided to adopt a criterion based on data available to the panel that were also included in our survey (age and gender): whenever we found a discrepancy in these characteristics, we deemed the answers unreliable and removed the participant from our dataset. Of course, there is no way to make sure that remaining data are indeed more reliable, as there isn’t in most of CAWI-based studies (that, however, is a widespread method for collecting data in sociology, psychology and social sciences). Nevertheless, we added a short statement in the “limitation” section.

10. While the authors stated the sample was representative, they did not provide comparison to the Italian adult population data, and the representativeness of the sample remains questionable.

Thank you for your feedback. In the table reporting sample characteristics we have added a column with comparative data. Moreover, in the text we added the link of the website of ISTAT (National Institute of Statistics) where we took the reference data. 

11. At line 182 the authors stated that “Ethical approval was obtained from the institutional review board at Catholic University of Milan (IRB#2019-12)”, that seemingly indicate the IRB occurred in December 2019, when nobody known about COVID-19 epidemic. Please could you address this issue, and provide the Ethical approval in the Supplement?

We thank the reviewer for this comment. We specified the ethical authorization process in the main text. The code IRB#2019-12 refers to the progressive number assigned to the protocol when submitted. We received the approval on January 2020. The study we report on is part of a larger continuative project which is aimed to monitor consumer behaviours along time on both fixed variables and ad hoc ones basing on the contingencies of the specific historical moment (i.e. the covid-19 pandemic in this case).

12. The idea to study the trust to authorities is very interesting, but the questions used seem questionable. Particularly, the statement “I fully trust the National Healthcare System” rated by on a scale ranging between 1 (completely disagree) to 5 (completely agree) is very non-specific, and could reflect not the trust to authorities but believes about financial stability of the NHS. This is a strong limitation, since it is among the central topics of the manuscript, and requires a caution reflection in the “Discussion”.

We partially agree with the reviewer’s concern. Due to time, space and methodological concerns, for the measurement of trust in authorities we decided to develop our own set of ad hoc items. Indeed, there is no way to be sure of how participants interpreted this particular statement (or the others, by the way) and we have added a statement in the limitations of this study. Nevertheless, the section of the survey in which there was this particular statement, was dedicated to Covid19 and Covid19-management (as pointed out in the instructions to participants), so we find it more likely that participants were oriented to interpret this question in this regard. Moreover, in Italy the NHS is a public institution, hence its financial stability is not generally a common concern.

13. Moreover, the question about media reliability (lines 228-231) is not related at all to the trust to authorities, and thus the whole analysis on measurement of trust to authorities become very questionable.

We agree with the reviewer that the question about media reliability is hardly related to the questions regarding trust in authorities. In trying to describe our study, we put it in the same section as we found that it was somewhat related to the topic. However, it is necessary to point out that for this very reason we never calculated a composite score out of these measures, and that they were treated as single dependent variables in the different analyses.

14. The question “have you reduced meals in ethnic restaurants?” at line 244 has no sense for persons who not visiting ethnic restaurants.

We agree with the reviewer’s concern: this was an oversight in the design of our questionnaire and hence we added a statement in the limitation section.

15 and 16. The description of questions regarding changes in purchasing behaviours at lines 250-260 is limited and does not contain the possible answer. As a general consideration, instead of describing all questions in the “Study measures” section, the authors could provide the complete questionnaire used during survey in the Supplement, and describe in the “Study measures” more precisely the meaning of the questions used.

Thank you for this suggestion, we have added our questionnaire as a supplement.

17. The presentation of the “Results” in Table 1, 3 and 4, 7 is not compliant for publication but resembles more the output from statistical software.

Tables have been revised.

18. Table 1 contains the description of the classification of participants according to the “Chronic patient” category, but any details about its definition are absent in the “Methods”.

Participants were simply asked if they were chronic patients or not. However, we agree that it need to be more explicitly stated in the methods section. Moreover, the exact question can now be found in the provided Supplement.

19. The “Partial Credit Rasch Model” described in “Results” should be described in the “Methods”, and of course it should be supported by the relevant references to the literature (that is insufficient in the manuscript now).

Thank you, we have added a few lines in paragraph 2.3.1 (lines 308-312) to better explain and reference the used model.

20. Tables should not include abbreviations in the title, or should include footer with their explanations.

Tables have been revised and abbreviations removed

21. The authors introduce some concepts “on the fly” in the “Results” without any explanations in the “Methods”, like “Person Separation Index”.

Thank you for you feedback: both the methods and the results sections have been revised. In particular, the Person Separation Index has been further explained at lines 432-435.

22. The calculation of the “Health Engagement Level“ is not clear, and not described in the manuscript. The authors should provide the exact explanations on which questions they used for defining one of the four possible levels. Without knowing how it was calculated, all subsequent discussion about the role of Health Engagement (that is a central idea of the manuscript) seems not having substantial ground.

The items of the Health Engagement scale are now visible in the Supplement. As for the scoring, this measure is available for use only with (free) licence and, as such, the Authors are not authorized to share details about scoring. 

23. The same concerns also all metrics mentioned in the Table 4. How the authors calculated “Risk severity”.

All the measures mentioned in Table 4 are single items with answers on a Likert scale ranging from 1 to 5 or 1 to 10

24. The description of differences between groups provided at lines 333-336 is repetitive. The same concerns lines 343-346.

25. Many lines that authors put in “Results” to describe statistical methods should be used in “Methods”.

26. Table 6 should not use plural for “logistic regression”, but most importantly should clearly define whether it was uni- or multivariable LR.

Thank you for your feedback. Both the Methods and Results sections have been revised to make them clearer and more organized. Tables also have been revised.

27. In the “Discussion” the authors stated that “many of Italian citizens experienced panic and enacted maladaptive behaviors”, without any examples that should be provided.

Thank you, discussion has been improved and some examples made explicit

28. The discussion is rather weak and includes rather few references to studies measuring impact of previous epidemics in other countries. This should be corrected.

Thank you for your feedback. The discussion of our manuscript has been revised, making particular attention to previous literature

29. “Limitations” should include issues pointed above.

The discussion of “Limitations” has been improved and expanded to include the issues you kindly pointed out. 

30. In the “Practical implications” the authors propose, for example to provide during the “ “psychological blackout”, reassuring messages, aimed at sustaining the emotional elaboration of the emergency and related worries would be particularly needed”. But they did not clarify what messages they exactly mean, and did not refer previously to any literature suggesting the content of such “reassuring messages”. The same concerns also other proposed by the authors stages of “Patient Health Engagement”. Thus, the practical implications remain rather vague.

Thanks for this suggestion. We further elaborate this part to improve the practical implication.

---

## [Decision Letter · Decision Letter 1]

12 Aug 2020

PONE-D-20-11202R1

Measuring Italian Citizens’ Engagement in the First Wave of the COVID-19 Pandemic Containment Measures: A Cross-sectional Study

PLOS ONE

Dear Dr. Barello,

Thank you for submitting your manuscript to PLOS ONE. After careful consideration, we feel that it has merit but does not fully meet PLOS ONE’s publication criteria as it currently stands. Therefore, we invite you to submit a revised version of the manuscript that addresses the points raised during the review process.

We look forward to receiving your revised manuscript.

Kind regards,

Wen-Jun Tu

Academic Editor

PLOS ONE

Reviewers' comments:

Reviewer's Responses to Questions

**Comments to the Author**

1. If the authors have adequately addressed your comments raised in a previous round of review and you feel that this manuscript is now acceptable for publication, you may indicate that here to bypass the “Comments to the Author” section, enter your conflict of interest statement in the “Confidential to Editor” section, and submit your "Accept" recommendation.

Reviewer #2: (No Response)

2. Is the manuscript technically sound, and do the data support the conclusions?

Reviewer #2: Partly

3. Has the statistical analysis been performed appropriately and rigorously? 

Reviewer #2: N/A

4. Have the authors made all data underlying the findings in their manuscript fully available?

Reviewer #2: No

5. Is the manuscript presented in an intelligible fashion and written in standard English?

Reviewer #2: No

6. Review Comments to the Author

Reviewer #2: In table1, chronic patient contained what types of disease and also the YES or No respectively represent the proportion of men and women or the proportion of chronic diseases in the total population?

In table3, the item was HE1,2,3,4,5,what does these mean?

In the part results, which contains too many methods,so please revise and reorganize the results and methods.

4.The following references should be discussed in the revision text.

Cao JL, Hu XR, Tu WJ., & Liu Q. (2020). Clinical Features and Short-term Outcomes of 18 Patients with Corona Virus Disease 2019 in Intensive Care Unit. Intensive Care Medicine, DOI: 10.1007/s00134-020- 05987-7.

Cao JL, Tu WJ, Hu XR, & Liu Q. (2020). Clinical Features and Short-term Outcomes of 102 Patients with Corona Virus Disease 2019 in Wuhan,China. Clinical Infectious Diseases,DOI: 10.1093/cid/ciaa243/ 5814897.

5.While the authors stated the sample was representative, they did not provide comparison to the Italian adult population data, and the representativeness of the sample remains questionable,So on behalf of the people who participated in the research, how to explain the reliability of the experiment

7. PLOS authors have the option to publish the peer review history of their article (what does this mean?). If published, this will include your full peer review and any attached files.

Reviewer #2: No

---

## [Author Response · Author response to Decision Letter 1]

19 Aug 2020

Comments from the editors and reviewer

We are very grateful to have been given the opportunity to revise our manuscript for Plos One.

We thank the referee and editor for their comments to strengthen the presentation of our work. We have modified the text to respond to all the issues and have elaborated on the changes below.

Comment 1

In table1, chronic patient contained what types of disease and also the YES or No respectively represent the proportion of men and women or the proportion of chronic diseases in the total population?

Response: 

Thanks for your comments. In our survey we asked participants whether they do or not suffer from a chronic disease; hence, the table 1 simply reports frequencies of responses (Yes vs No) in our sample, we have no means to know the types of diseases of which they suffer. We changed the label for more clarity.

Comment 2

In table3, the item was HE1,2,3,4,5,what does these mean?

Response:

Thanks for this comment. It means “health engagement”, which is the measured variable. We modified the label for more clarity.

Comment 3

In the part results, which contains too many methods, so please revise and reorganize the results and methods.

Response:

Thanks. We revised the two sections moving some part from the results to the method section.

Comment 4

The following references should be discussed in the revision text.

Cao JL, Hu XR, Tu WJ., & Liu Q. (2020). Clinical Features and Short-term Outcomes of 18 Patients with Corona Virus Disease 2019 in Intensive Care Unit. Intensive Care Medicine, DOI: 10.1007/s00134-020- 05987-7.

Cao JL, Tu WJ, Hu XR, & Liu Q. (2020). Clinical Features and Short-term Outcomes of 102 Patients with Corona Virus Disease 2019 in Wuhan,China. Clinical Infectious Diseases, DOI: 10.1093/cid/ciaa243/ 5814897.

Response:

Thanks for this suggestion. We added this reference in the introduction section.

Comment 5

While the authors stated the sample was representative, they did not provide comparison to the Italian adult population data, and the representativeness of the sample remains questionable, So on behalf of the people who participated in the research, how to explain the reliability of the experiment

Response:

The sample was representative of the Italian population due to the sampling method adopted, we’ve added further details in the methods section. Comparison data are provided in Table 1 under the “% Italian population” columns and retrieved from the website of ISTAT (the Italian National Institute of Statistics). Therefore, we assume that the study results are valid for not only for the sample of the study, but also for the larger Italian Population. Surely, further study on other sample should confirm that. We stressed this aspect in the limitation section.

---

## [Editor Report · Decision Letter 2]

21 Aug 2020

Measuring Italian Citizens’ Engagement in the First Wave of the COVID-19 Pandemic Containment Measures: A Cross-sectional Study

PONE-D-20-11202R2

Dear Dr. Barello,

We’re pleased to inform you that your manuscript has been judged scientifically suitable for publication and will be formally accepted for publication once it meets all outstanding technical requirements.

Kind regards,

Wen-Jun Tu

Academic Editor

PLOS ONE
---

## [Editor Report · Acceptance letter]

28 Aug 2020

PONE-D-20-11202R2 

Measuring Italian Citizens’ Engagement in the First Wave of the COVID-19 Pandemic Containment Measures: A Cross-sectional Study 

Dear Dr. Barello:

I'm pleased to inform you that your manuscript has been deemed suitable for publication in PLOS ONE. Congratulations! Your manuscript is now with our production department. 

Kind regards, 

on behalf of

Dr. Wen-Jun Tu 

Academic Editor

PLOS ONE